# Systematic review of ecological momentary assessment (EMA) studies of five public health-related behaviours: review protocol

Dominika Kwasnicka [ID],[1,2] Dimitra Kale [ID],[3] Verena Schneider [ID],[3]
Jan Keller [ID],[4] Bernard Yeboah-Asiamah Asare [ID],[5,6] Daniel Powell [ID],[6,7]
Felix Naughton [ID],[8] Gill A ten Hoor [ID],[9] Peter Verboon [ID],[10] Olga Perski [ID][3]

For numbered affiliations see end of article.

**Correspondence to**
Dr Dominika Kwasnicka;
dkwasnicka@swps.edu.pl

## ABSTRACT

**Introduction** Ecological momentary assessment (EMA) involves repeated, real-time assessments of phenomena (eg, cognitions, emotions, behaviours) over a period of time in naturalistic settings. EMA is increasingly used to study both within-person and between-person processes. We will review EMA studies investigating key health behaviours and synthesise: (1) study characteristics (eg, frequency of assessments, adherence, incentives), (2) associations between psychological predictors and behaviours and (3) moderators of adherence to EMA protocols.

**Methods and analysis** This review will focus on EMA studies conducted across five public health behaviours in adult, non-clinical populations: movement behaviour (including physical activity and sedentary behaviour), dietary behaviour, alcohol consumption, tobacco smoking and preventive sexual health behaviours. Studies need to have assessed at least one psychological or contextual predictor of these behaviours. Studies reporting exclusively on physiological outcomes (eg, cortisol) or those not conducted under free-living conditions will be excluded. We will search OVID MEDLINE, Embase, PsycINFO and Web of Science using terms relevant to EMA and the selected health behaviours. Reference lists of existing systematic reviews of EMA studies will be hand searched. Identified articles will be screened by two reviewers. This review is expected to provide a comprehensive summary of EMA studies assessing psychological or contextual predictors of five public health behaviours.

**Ethics and dissemination** The results will be disseminated through peer-reviewed publications and presentations. Data from included studies will be made available to other researchers. No ethics are required.

**PROSPERO registration number** CRD42020168314.

## Strengths and limitations of this study

► A protocol for a systematic review is provided for ecological momentary assessment (EMA) studies in adult, non-clinical populations.
► We will include EMA studies of five key public health behaviours including movement behaviours, dietary behaviours, alcohol consumption, tobacco smoking and preventive sexual health behaviours.
► We will review characteristics of EMA studies (eg, study duration in days, incentives, adherence rates) and associations between psychological predictors and behaviours, examining rates of adherence to EMA protocols across different health behaviours and moderators of adherence (eg, study setting, type of behaviour).
► Extracted data will be made available to other researchers, thus allowing for the exploration of additional research questions and potential for setting up a 'living review'.
► As included studies are likely to be heterogeneous, this may limit the overarching conclusions that can be drawn, and will likely prevent meta-analysis combining effect sizes from multiple studies and across all behaviours.

EMA is increasingly used to study within-person and between-person processes, including associations between psychological and health behaviour-related variables (eg, positive affect and physical activity in general population samples or stress and lapse risk in smokers attempting to stop). For researchers and healthcare professionals to understand and change behaviour, it is important for theories and interventions to be applicable to both momentary states of individuals (within-person processes) and groups of individuals (between-person processes).[2] Despite the popularity and importance of EMA for studying health-related behaviours, there has been no comprehensive systematic

## INTRODUCTION

Ecological momentary assessment (EMA), also known as ambulatory assessment or experience sampling methodology, involves repeated, real-time assessments of phenomena (eg, cognitions, emotions and behaviours) over a period of time in naturalistic settings.[1]

investigation of characteristics of EMA studies (eg, rates of adherence, study duration in days, incentive schedules) and potential moderators of adherence (eg, study setting, type of health behaviour), with attempts to describe associations between psychological predictors (eg, intentions, self-efficacy) and key public health behaviours.

Previous reviews of EMA studies have focused on clinical conditions such as borderline personality disorder,[3] psychotic disorder,[4] mood disorders,[5] binge eating,[6] bulimia nervosa,[7] anxiety disorder,[8] schizophrenia,[9] alcohol use disorder,[10] chronic pain[11] and specific populations such as children and adolescents,[12] youth[13] and older adults.[14] Health behaviour-specific reviews of EMA studies have focused on physical activity,[15 16] sedentary behaviour,[16] alcohol use,[17] craving and substance use,[18] dietary behaviours[19] and the relationship between alcohol use and sexual decision-making.[20] Previous EMA reviews have also focused on inter-relations between specific psychological variables, such as the association of everyday social interactions with intra-individual variability in affect.[21]

While systematic reviews of EMA studies focusing on specific health behaviours have been conducted,[18 22 23] there are no overarching reviews that can help address broad questions about study characteristics (eg, study duration in days, adherence, incentive schedules), rates and moderators of adherence (eg, study setting, participant characteristics) and predictor-behaviour associations across different health behaviours. It is expected that this review will help fill this gap. We also expect that this review will help inform the design of future EMA studies by providing a summary of best practice across research contexts, settings and health-related behaviours. For instance, results may be useful for informing researchers' understanding of what frequency or intensity of change we would expect to see at what temporal resolution (ie, informed by a 'theory of change'[24]), which can then inform assessment scheduling decisions. This review is likely to include a large number of studies, thus providing a comprehensive overview of the EMA literature.

### The current study
We will synthesise evidence from EMA studies that report either within-person or between-person predictor-behaviour associations. The review will focus on five key public health behaviours: (1) movement behaviours (including physical activity and sedentary behaviour), (2) dietary behaviours, (3) alcohol consumption, (4) tobacco smoking and (5) preventive sexual health behaviours (including contraceptive use).

The review aims are:
1. To summarise adherence to EMAs, total length of data collection of EMAs, prompting frequency of EMAs, and incentives structures across studies.
2. To describe within-person and between-person predictor-behaviour associations across EMA studies (eg, associations between intention and behaviour).

3. To assess potential moderators of adherence to EMAs (eg, study setting, participant characteristics).

This review is intentionally broad in scope to provide an overview of the field for researchers interested in the application of EMAs to the study of health-related behaviours. We expect this overarching review to help identify patterns and key knowledge gaps.

## METHODS AND ANALYSIS
### Study design
This review will adhere to the Preferred Reporting Items for Systematic Reviews and Meta-Analyses (PRISMA) checklist (online supplemental material 1). The review start date was 15 September 2019 and the planned end date is 30 December 2021.

### Inclusion criteria
This review will focus on five key public health behaviours in healthy adults (ie, non-clinical populations) aged 18+years, namely:
1. Movement behaviours, including physical activity and sedentary behaviour.
2. Dietary behaviours, including snacking or fruit and vegetable consumption.
3. Alcohol consumption.
4. Tobacco smoking, including cigarette, cigar or pipe smoking.
5. Preventive sexual health behaviours, including contraceptive/condom use.

No restrictions on geographical location or publication date will be set. To be included, studies need to incorporate multiple (ie, two or more) within-day, daily or weekly assessments of predictors and behaviours, and to have reported either (or both) within-person or between-person predictor-behaviour (eg, stress predicting unhealthy snack consumption) associations. The frequency of the EMAs should plausibly match how the target behaviour (and psychological and contextual predictors) theoretically or empirically unfolds over time, for example, daily assessments of steps, weekly assessments of gym class attendance if the class is undertaken only once a week. To be included, studies need to assess one of the aforementioned behaviours and at least one psychological or contextual variable via EMAs.

In this review, we defined psychological variables as emergent properties of a distributed network of neurons, including cognition (eg, beliefs, attitudes, goals), emotion (eg, negative affect, cravings) and processes operating on these (eg, self-regulation, learning), which are linked to behaviour. We further define contextual variables as any potential environmental (eg, social or physical) influences on behaviour, including the presence of other people, weather or the availability of unhealthy foods/cigarettes/alcohol. The psychological and contextual variables will be closely assessed by the reviewers as to their suitability for inclusion/exclusion in the review.

In addition to self-report measures, included studies can use physiological measures of psychological predictors (eg, cortisol or heart rate variability to measure stress) or behaviours (eg, accelerometer data to measure physical activity or sedentary behaviour). Studies reporting associations between behaviours and psychological consequences (eg, whether physical activity predicts affect) will be included providing that they also report psychological or contextual predictor-behaviour associations (eg, whether positive affect predicts physical activity). We will include individuals with overweight and obesity given that 39% of adults globally fall into this category, with most Western countries averaging above 50%.[25] Studies including participants with a diagnosed mental or physical health condition who were not recruited into the study on the basis of their condition will be included (eg, studies including participants with clinical levels of depression but where this was not an inclusion criterion). Studies in which a behavioural or pharmacological intervention was delivered will be included providing that participants were asked to complete free-living EMAs.

## Exclusion criteria

Laboratory studies will not be included. Studies examining clinical populations, that is, solely recruiting participants on the basis of being diagnosed with a physical or mental health condition such as cancer, cardiovascular disease, depression, binge eating disorder or substance use disorder (also including case–control studies) will be excluded. Studies focusing only on purchasing behaviours (eg, tobacco purchasing, food purchasing) will not be included. Studies focusing on e-cigarettes will be also excluded. Studies not published in English or where no full text could be obtained will also not be included. Although behaviour–behaviour associations may also be considered relevant, our electronic search is not designed to capture such studies, and behaviour–behaviour associations will hence not be considered further in this review.

## Search methods for the identification of studies
### Electronic searches

We will search Ovid MEDLINE, Embase, PsycINFO and Web of Science (see online supplemental material 2 for the full search strategy). Terms will be searched in titles and abstracts as free-text terms or as index terms (eg, Medical Subject Headings), as appropriate. We will combine two groups of terms, the first with terms relevant to EMAs and within-person study designs; the second with terms relevant to the five health behaviours addressed in this review.

Example terms used:
1. (ecological adj1 momentary adj1 assessment*) OR (intensive adj1 longitudinal) OR (ambulatory adj1 assessment*) OR (experience adj1 sampl*) OR (daily adj1 diar*) OR (ecological adj1 momentary adj1 intervention) OR within-person OR within-subject* OR (single adj1 case) OR idiographic OR intraindividual

2. tobacco OR smok* OR cigarette OR alcohol* OR drinking OR addict* OR (healthy adj3 eat*) OR diet OR weight OR overweight OR obes* OR physical activity OR exercise OR sedentary OR sitting OR leisure OR (sexual adj1 health) OR condom OR contraceptive
3. 1 AND 2

Electronic and hand searches were conducted in January 2020 and updated in February 2021. We restricted the search to human studies available in English that are published in peer-reviewed journals (online supplemental material 2).

### Searching for other sources

Reference lists of existing systematic reviews of EMA studies will be hand searched and expertise within the review team will be used to identify additional articles of interest.

## Data collection and analysis
### Selection of studies

Identified articles will be merged using Covidence[26] and duplicate records will be removed. The three lead authors (DKw, OP and JK) will independently screen titles and abstracts (yes, maybe, no) against the pre-specified inclusion criteria. Full texts will be screened by two reviewers independently (yes, no); discrepancies will be resolved by the lead authors and inclusion will be further discussed with other team members if needed. In line with the PRISMA checklist, key reasons for exclusion will be recorded at the full-text stage. These will include: lack of psychological predictors or outcomes; study not being relevant to the five key public health behaviours of interest; wrong study design (not an EMA study); participants being recruited based on a health condition (ie, clinical population); participants younger than 18 years old; studies of purchasing behaviours; conference abstracts; protocols; duplicates; studies not published in English or no full text could be obtained. We will follow the hierarchy of the exclusion criteria, listing the first reason from the aforementioned list as the key reason for exclusion.

### Data extraction and management

A data extraction form will be developed in Microsoft Excel to extract information and to import data into R for analysis. Each study will be allocated a unique study identification number. Data will be extracted on:
- *Study description* (study author, year, country, study funder);
- *Participant characteristics* (sample size; mean or median age (SD); gender (% female); educational attainment (% university education); population type (eg, men who have sex with men, older adults, general population), ethnicity (% white ethnicity);
- *EMA study type* (eg, observational, interventional, both);
- *EMA delivery mode* (eg, mobile phone, website/online, pen-and-paper);

- *EMA method* (eg, signal contingent, event contingent, multiple);
- *EMA characteristics* (eg, total study duration in days); prompting frequency (eg, hourly, daily, weekly), incentive schedule (eg, flat rate, payment per EMA);
- *Adherence to EMA* (eg, average % EMAs completed out of available prompts);
- *Health behaviour(s) assessed* (eg, physical activity, sedentary behaviour, dietary behaviour, tobacco smoking); and how the health behaviour(s) were measured (eg, hourly step count, number of cigarettes smoked per day);
- *Psychological and contextual predictors* (eg, intentions, self-efficacy, presence of other smokers) and how they were measured (eg, EMA method, measurement frequency, whether the measure was developed for the study (vs there being a precedent for the measure), whether a single item or multiple items were used);
- *Statistical model used to examine predictor-behaviour association* (eg, multilevel model, generalised estimating equation) and whether these associations were analysed at the within-person and/or between-person level;
- *Level of aggregation in data analysis* (ie, whether data underpinning the predictor-behaviour association are aggregated vs maintained at the within-person level);
- *Coefficients and effect sizes from statistical models* (eg, ORs, relative risks, regression coefficients);
- *Control variables in multivariable models* (eg, age, sex)

For each study, one reviewer will extract the data. At least 20% of studies stratified by behaviour (eg, 20% of all alcohol consumption studies) will be double-checked for accuracy and completeness by a second reviewer. In case there are any uncertainties related to data extraction (eg, the primary data extractor is uncertain about a particular parameter or a large number of discrepancies are observed across the primary and secondary data extractor), we will double-check additional studies until agreement is achieved. All review authors will be involved in data extraction and double-checking.

### Quality appraisal

Included studies may vary in quality, which will be considered through a quality appraisal. The appraisal tool was developed by the review team, based on an existing EMA reporting checklist,[27] and includes the following four criteria: (1) rationale for EMA design, (2) a priori power analysis to determine sample size, (3) percentage adherence to the EMA protocol and (4) treatment of missingness (table 1). The quality indicators will be coded by one reviewer, with 20% or more double-checked by a second reviewer. Discrepancies will be resolved through discussion among the lead authors. As each criterion refers to a different aspect of study quality, we will not summarise study quality, but will present how studies score on each selected dimension.

### Data synthesis

All quantitative analyses will be conducted in R V.3.5.1. A narrative (descriptive) synthesis will be conducted. We will summarise the number of EMA studies conducted for each of the five health behaviours, study setting (eg, country, immediate study setting) and sample size (ie, mean or median number of participants per study). We

| Table 1 | Quality appraisal of included EMA studies | | |
|---|---|---|---|
| **Topic: quality criteria** | **Strong** | **Moderate** | **Weak** |
| **Rationale** | | | |
| **1. Rationale for EMA design provided**: Why was an EMA design chosen to examine the research question? | A strong rationale provided for the EMA design of predictor AND behaviour/ outcome. | Rationale provided but not very strong for the EMA design of either the predictor OR behaviour/ outcome. | No rationale for the EMA design regarding predictor and behaviour/outcome. |
| **Power analysis, sample size and participant adherence** | | | |
| **2. Power analysis**: A priori power analysis to determine sample size | An a priori power analysis is reported and the enrolled sample size met power analysis indication / OR: sufficient explanation as to why an a priori power analysis was not needed | An a priori power analysis is reported but sufficient sample size/ number of observations was not achieved. | No information about power analysis / OR: a post-hoc power analysis is reported. |
| **3. Adherence to EMA protocol:** Percentage of answered EMA prompts across all participants for the main EMA study period | Percentage of answered EMA prompts >80%. | Percentage of answered EMA prompts 60%–79.99%. | Percentage of answered EMA prompts <60%. |
| **Data analysis** | | | |
| **4. Treatment of missingness:** Report whether study dropout or non-adherence to EMAs (eg, missed prompts) are related to specific variables | Missing mechanisms/predictors are identified, reported and mitigated for if needed. | Missing mechanisms/predictors are identified and reported but not mitigated for. | Missing mechanisms/predictors are not identified or reported. |

EMA, ecological momentary assessment.

will then present results in relation to each research question.

To address the first aim, we will summarise study and EMA characteristics, for example, study setting, population characteristics, percentage prompting frequency (eg, % daily, % weekly), percentage type of EMA method (eg, % event contingent, % random assessments, % continuous sensor based, % hybrid), percentage type of EMA delivery mode (eg, % smartphone application delivery), percentage type of incentive structure (eg, % flat payment, % payment per EMA, % no incentive), rates of EMA adherence (mean or median), and study duration (mean or median). Where appropriate, moderator analyses will be conducted to examine whether predictor-behaviour associations vary depending on study setting, study characteristics, participant characteristics, or type of incentive schedule used.

To address the second aim, we will summarise within-person and between-person predictor-behaviour associations across EMA studies (eg, the type of psychological or contextual predictor assessed, measurement type, frequency of measurement). If there is sufficient homogeneity between studies (eg, similar predictors assessed with similar measurement type and frequency across ≥3 studies), within-person or between-person predictor-behaviour associations (eg, ORs, relative risks, regression coefficients) will be synthesised with random effects meta-analyses, grouped by behaviour. Analyses will be conducted with the 'metafor' or 'CTmeta' packages,[28–30] as appropriate, also using 'jamovi'.[31] Where sufficient detail on model parameter estimates is lacking in the publications, we may contact study authors to request access to additional information.

To address the third aim, we will assess, with random effects meta-analyses, whether EMA adherence varies depending on study setting, study characteristics, participant characteristics or type of incentive schedule used. We do not have any pre-specified hypotheses.

### Patient and public involvement

A patient and public involvement representative reviewed a lay summary of the protocol for our systematic review. Positive feedback was received on the review's aims, the importance of the current research and choice of key behaviours relevant to public health. Once the review is completed, feedback will be sought from additional patient and public involvement representatives about the interpretation of findings and plans for dissemination. We will seek advice on how to best present the study outcomes and use them in order to design studies and interventions that are useful and relevant for the public.

### ETHICS AND DISSEMINATION

This study does not require ethics approval as it will summarise data from previously published studies. A protocol was pre-registered on the international Prospective Register of Systematic Reviews and on the Open Science Framework. The findings of the review will be disseminated through peer-reviewed publications and presentations at relevant conferences. The data set will be made available to other researchers online via the creation of a digital object identifier, thus enabling further research questions to be addressed. We expect this review to be useful for researchers and healthcare practitioners who regularly design and interpret results from EMA studies. We plan to publish overarching review and subsequently five behaviour-specific reviews that will provide a more in-depth synthesis of predictor-behaviour associations.

### Summary

EMA is a frequently used research method; however, an overview of studies using this method across key public health behaviours in healthy adults is lacking. This review will provide a comprehensive overview of associations between a psychological/contextual predictor and a health behaviour in EMA studies focusing on movement behaviours, dietary behaviours, alcohol consumption, tobacco smoking and sexual health behaviours. This review will inform the future design of EMA studies and it will influence practice of assessing individuals in real-life settings and providing interventions that are delivered at the time and place when and where required. This review will set a blueprint for how to conduct EMA studies to improve participants' adherence and conduct meaningful studies in real-life settings.

### Author affiliations
[1]Faculty of Psychology, SWPS University of Social Sciences and Humanities, Wroclaw, Poland
[2]NHMRC CRE in Digital Technology to Transform Chronic Disease Outcomes, Melbourne School of Population and Global Health, University of Melbourne, Melbourne, Victoria, Australia
[3]Department of Behavioural Science and Health, Institute of Epidemiology and Health Care, Faculty of Population Health Sciences, University College London, London, UK
[4]Department of Education and Psychology, Freie Universität Berlin, Berlin, Germany
[5]Curtin School of Population Health, Curtin University, Perth, Western Australia, Australia
[6]Health Psychology, Institute of Applied Health Sciences, University of Aberdeen, Aberdeen, UK
[7]Rowett Institute, University of Aberdeen, Aberdeen, UK
[8]Behavioural and Implementation Science Group, School of Health Sciences, University of East Anglia, Norwich, UK
[9]Department of Work and Social Psychology, Faculty of Psychology and Neurosciences, Maastricht University, Maastricht, The Netherlands
[10]Faculty of Psychology, Open University, Heerlen, The Netherlands

**Acknowledgements** The review team would like to thank Dr Marta Marques for comments and suggestions on the initial version of the review protocol. The authors would like to thank the patient and public involvement representative who commented on the lay summary of our proposed plan, for their contribution to this research.

**Contributors** DKw, OP, DP and FN conceived the project. DKw and OP are the project leads and coordinators, they jointly drafted the manuscript. All authors (DKw, DKa, VS, JK, BY-AA, DP, FN, GAtH, PV and OP) have made conceptual contributions to project design and procedures. All authors read, edited and approved the final version.

**Funding** DKw's work is carried out within the HOMING programme of the Foundation for Polish Science co-financed by the European Union under the European Regional Development Fund; grant number POIR.04.04.00-00-5CF3/18-00; HOMING 5/2018. DKa and OP receive salary support from Cancer Research UK (C1417/A22962). DP is funded by the Scottish Government's Rural and Environment Science and Analytical Services (RESAS) and by the School of Medicine, Medical Sciences, and Nutrition (SMMSN) at the University of Aberdeen. FN's salary is covered by the Faculty of Medicine and Health Sciences at the University of East Anglia.

**Competing interests** None declared.

**Patient consent for publication** Not required.

**Provenance and peer review** Not commissioned; externally peer reviewed.

**ORCID iDs**
Dominika Kwasnicka http://orcid.org/0000-0002-5961-837X
Dimitra Kale http://orcid.org/0000-0002-8845-7114
Verena Schneider http://orcid.org/0000-0003-0244-6201
Jan Keller http://orcid.org/0000-0003-4660-6844
Bernard Yeboah-Asiamah Asare http://orcid.org/0000-0002-1381-4981
Daniel Powell http://orcid.org/0000-0003-4995-6057
Felix Naughton http://orcid.org/0000-0001-9790-2796
Gill A ten Hoor http://orcid.org/0000-0001-5500-1893
Peter Verboon http://orcid.org/0000-0001-8656-0890
Olga Perski http://orcid.org/0000-0003-3285-3174

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
