## [Reviewer comments · BMJ Open]

ARTICLE DETAILS

TITLE (PROVISIONAL)	Systematic review of Ecological Momentary Assessment (EMA) studies of five public health-related behaviours: Review Protocol.
AUTHORS	Kwasnicka, Dominika; Kale, Dimitra; Schneider, Verena; Keller, Jan; Yeboah-Asiamah Asare, Bernard; Powell, Daniel; Naughton, Felix; ten Hoor, Gill A.; Verboon, Peter; Perski, Olga

VERSION 1 – REVIEW

REVIEWER	Romanzini, Catiana State University of Londrina, Department of Physical Education
REVIEW RETURNED	21-Jan-2021

GENERAL COMMENTS	The paper is well written and conducted like the proposal by BMJ Open. The paper doesn't expose the studies. My only suggestion for the authors is chose: 1) add another health behavior (sedentary behaviour) in the entire paper (title, abstract, introduction and methods) or 2) excluded this. My appointment in relation to this suggestion are because the important researchers in field of physical activity and sedentary behaviour, have already demonstrated that these two behaviors are distinct of each other (TREMBLAY et al., 2017). Other considerations:  • In abstract the author cited only 'physical activity' (page 6; line24); • In introductions the authors cited two studies with EMA and physical activity and one study with sedentary behaviour; this demonstrated that they are distinct (page 8; line 52); • In method the authors cited that physical activity (including sedentary behaviour) (page 9; line 45 and page 10; line 29); • In summary the authors cited 'on physical activity and sedentary behaviour' (page 17; line 47); • In supplementary material 2 the authors don't cite any keyword related to 'sedentary behaviour'; only 'physical activity' and 'exercise'; • In protocol registered with PROSPERO, available in www.crd.york.ac.uk/prospero/display_record.php?ID=CRD42020168314, the only keyword used by authors is 'sedentary time', that are different that 'sedentary behaviour', that use in this paper. There are many forms of sedentary behaviour that occurring all day long (24hs movement). And finally, according 'sedentary behaviour' definition, is "any waking behavior characterized by an energy expenditure ≤ 1.5 metabolic equivalents (METs), while in a sitting, reclining or lying posture", the search only with keyword 'physical activity' and 'exercise' don't be sufficient to cover the literature about
--

	'sedentary behaviour' (TREMBLAY et al., 2017, p.9). Tremblay, M.S., Aubert, S., Barnes, J.D. et al. Sedentary Behavior Research Network (SBRN) – Terminology Consensus Project process and outcome. Int J Behav Nutr Phys Act 14, 75 (2017). https://doi.org/10.1186/s12966-017-0525-8 .
--	---

REVIEWER	Maier, Jaclyn P. Univ N Carolina
REVIEW RETURNED	11-Feb-2021

GENERAL COMMENTS	This paper is a review protocol and outlines procedures for conducting a review of ecological momentary assessment studies of five distinct health behaviors. My main concern with this protocol is how broad the review is. There are 5 distinct health behaviors which are to be included and the overall objective aim to look at three distinct facets: 1. details of EMA protocol and compliance, 2. associations between psychological factors and behavior (in both directions), and 3. moderators of those associations. I worry about the potential to provide a coherent, digestible review given the broad nature of what is proposed. I have additional, more specific comments outlined below. 1. How are the authors defining psychological constructs? At various points through the protocol, they provide examples such as affect, intentions, self-efficacy, but they do not ever seem to actually define the term. Without this it would seem difficult to be able to replicate or determine whether all applicable articles had been included. 2. The authors acknowledge that the topic of physical activity will include sedentary behavior, but they have no search terms dedicated to that behavior. Given that physical activity and sedentary behavior are distinct behaviors, it seems as though it would be important to integrate this (and other relevant) search terms. 3. What is the rationale for including weekly assessments as within the umbrella of EMA? Certainly for physical activity and sedentary behavior, health behaviors with occur multiple times per day, it would seem that weekly assessments would be extrapolating too far for the occasions when these behaviors actually occur (which is the purpose of EMA). Are there certain health behaviors where you would consider weekly assessments to be real time, but others not? 4. The data extraction section doesn't seem to list any information related to objectives 2 or 3. What is listed seems to exclusively focus on describing study characteristics and methods as opposed to the associations examined within these studies.
--

REVIEWER	Yang, Chih-Hsiang University of South Carolina
REVIEW RETURNED	13-Feb-2021

GENERAL COMMENTS	This study provides a protocol for reviewing characteristics and features in EMA studies targeting five different health behaviour outcomes. As the EMA literature is growing rapidly in the public health field, synthesizes work is critical and time-sensitive to guide and advance future studies. I consider this proposed work will provide a significant contribution to the field. The study protocol
---

overall is rigorous, and it complies with current recommendations for conducting systematic reviews. There are some questions and clarifications needed in the current version of the manuscript. I provided my comments and questions for each section for the authors' reference.

Abstract

- Lines 26-28. "Studies need to have assessed at least one psychological or contextual predictor of these behaviours." The inclusion of at least one "contextual predictor" may be too broad for the scope of this review. Contextual factor can also be defined as physical and social contexts, aside from psychological context.
- Lines 31 -33. The authors mentioned that "reporting exclusively on physiological outcomes (e.g., cortisol) or not conducted under free-living conditions will be excluded." Cortisol could also be a psychological outcome to measure stress level. Would it be excluded if a study measures health behaviors and cortisol based on this criterion?
- p.4. Line 16. "length of follow-up", and "assessment type" is unclear. Do the authors mean the time intervals between EMA assessment? And what are some examples for assessment type?

Introduction

- p.6. A brief literature review/summary of EMA adherence should also be included in the intro.
- p.6. Line 45. "healthy eating" was only mentioned once throughout the MS. Dietary behavior may be a better and more comprehensive term.
- p.7. Lines 3-5. "predictor-behaviour" and "predictor-outcome" is not clear. Could the authors provide examples to distinguish the two?
- p.7. Lines 36. Is e-cigarette use included in this review?
- p.7. Lines 36. Again, including at least one contextual factor may not help narrow the scope of this review.
- p.9. Some critical terms are missing in the search terms. For example, "intraindividual", "drinking", "cigarette", "movement behaviours", "leisure activity", "sedentary behaviours", "addictive behaviour."

Data extraction

- p.10. Line 54. Examples of psychological predictors (e.g., intentions, self-efficacy) should be brought up and define well earlier in the intro as they are key variables in this review.
- p.10. Line 26 "A data extraction form will be developed in Microsoft Excel by the two lead authors" and p.11. Line 18 "For each study, one reviewer will extract the data. Twenty percent of studies will be double checked for accuracy and completeness by a second reviewer." Could you clarify if the two reviewers are the same as the two lead authors?
- I highly suggest that how the associations between psychological predictors and the behavioural outcome should also be synthesized. For example, does the study only calculated correlation or use other modeling approaches to test the associations. It will make a valuable contribution to the field.
- Is there a rationale for only selecting 20% of the studies for double-checking? Some health behavior categories also have more EMA studies than other fields, so the probability for studies being selected for a double check may not be consistent across health behaviours.
- p. 13. Lines 22. Standard regression analyses may not be feasible for categorical variables such as setting type, characteristics. Please clarify what analysis will be used for these.

Summary

- p. 56. "participant burden" was mentioned the first time in the last

	sentence of the MS. Study burden is another topic that should not be mixed with study compliance in EMA studies. If reducing burden is another focus of this review to inform future studies, then more discussions are needed in the intro.
--	--

REVIEWER	Shiffman, Saul University of Pittsburgh, Psychology
REVIEW RETURNED	18-Feb-2021

GENERAL COMMENTS	This is a reposed protocol for conducting a systematic review of studies of health-relevant behaviors using EMA methods. EMA methods have come into very widespread use in the research areas targeted, and the field could benefit from systematic reviews, so the proposed paper would address a need. The proposal is generally solid and well-thought-out, but there are a few issues that could be better addressed. The search terms seem appropriate, and reflect a knowledge of the variety of traditions in which such research is conducted. The scope is rather too wide, particularly in trying to encompass studies with multiple within-day measures – the area where innovation has been occurring – along with measures taken only daily or weekly. Especially including weekly measures seems very far outside the scope, and would seem to include studies dating back to the 1800s. At best, this would dilute the review; at worst, the results could be quite muddled. It is recommended the review focus on the more frequent measures that are relatively new. In another area, the review is too narrow. It proposes to focus only on healthy adults. But a lot of this work is occurring in clinical populations, and it would limit the review to exclude those. Moreover, defining the boundaries is bound to be problematic: are 'problem drinkers' a clinical population or healthy adults with drinking problems? They propose to include studies of obese individuals: Are such individuals 'healthy' or 'clinical'? It may be wise to exclude small highly specialized studies, but not wise to exclude all 'clinical' populations. The 'quality' appraisal needs some revision. It mixes (a) factors that reflect the quality of the DATA (e.g., degree of adherence, which may reflect bias or lack thereof); (b) factors that reflect the quality of REPORTING, such as whether a rationale is stated for the EMA design; and (c) factors that are not 'quality' at all, such as the degree of aggregation, where almost any degree can be appropriate if it fits the data and hypothesis. It seems especially unfortunate to aggregate these diverse considerations into a single index... it could be very misleading. In sum, the focus and methods seem very sound, but revisions in scope and scoring have the potential to materially improve the resulting review.
---

VERSION 1 – AUTHOR RESPONSE

Reviewer: 1

Dr. Catiana Romanzini, State University of Londrina

Comments to the Author:

The paper is well written and conducted like the proposal by BMJ Open. The paper doesn't expose the studies.

Response: We would like to thank the reviewer for their positive feedback on our review protocol.

My only suggestion for the authors is chose: 1) add another health behavior (sedentary behaviour) in the entire paper (title, abstract, introduction and methods) or 2) excluded this. My appointment in relation to this suggestion are because the important researchers in field of physical activity and sedentary behaviour, have already demonstrated that these two behaviors are distinct of each other (TREMBLAY et al., 2017).

Response: We agree that physical activity and sedentary behaviour are two distinct behaviours and that it would be important to better account for the latter. We have therefore made the following amendments to the methods and search strategy: we have added the terms 'sedentary', 'sitting' and 'leisure' to our search. We have also justified in text that because movement behaviour is often perceived on a continuum from being sedentary to active, we included studies exploring sedentary behaviour as part of a broader movement behaviour category (changes made throughout the article).

Other considerations:

- In abstract the author cited only 'physical activity' (page 6; line24);

Response: We have altered the abstract to reflect this suggestion: "This review will focus on EMA studies conducted across five public health behaviours in adult, non-clinical populations: movement behaviour (including physical activity and sedentary behaviour), dietary behaviour, alcohol consumption, tobacco smoking, and preventive sexual health behaviours."

- In introductions the authors cited two studies with EMA and physical activity and one study with sedentary behaviour; this demonstrated that they are distinct (page 8; line 52);

Response: We fully agree and we have now implemented aforementioned changes.

- In method the authors cited that physical activity (including sedentary behaviour) (page 9; line 45 and page 10; line 29);

Response: We fully agree and we implemented aforementioned changes.

- In summary the authors cited 'on physical activity and sedentary behaviour' (page 17; line 47);

Response: We fully agree and we kept it as that fits with new revised wording.

- In supplementary material 2 the authors don't cite any keyword related to 'sedentary behaviour'; only 'physical activity' and 'exercise';

Response: We have added such keywords (e.g., ' sedentary' and 'sitting').

- In protocol registered with PROSPERO, available in www.crd.york.ac.uk/prospero/display_record.php?ID=CRD42020168314, the only keyword used by authors is 'sedentary time', that are different that 'sedentary behaviour', that use in this paper.

Response: We have submitted changes to the protocol registered online on PROSPERO to reflect the updated search. PROSPERO may take some time to process the changes and they may therefore not yet be included online when the article is resubmitted.

There are many forms of sedentary behaviour that occurring all day long (24hs movement). And

finally, according 'sedentary behaviour' definition, is "any waking behavior characterized by an energy expenditure ≤ 1.5 metabolic equivalents (METs), while in a sitting, reclining or lying posture", the search only with keyword 'physical activity' and 'exercise' don't be sufficient to cover the literature about 'sedentary behaviour' (TREMBLAY et al., 2017, p.9).

Tremblay, M.S., Aubert, S., Barnes, J.D. et al. Sedentary Behavior Research Network (SBRN) – Terminology Consensus Project process and outcome. *Int J Behav Nutr Phys Act* 14, 75 (2017). <https://doi.org/10.1186/s12966-017-0525-8>.

Response: We fully agree. This is a very relevant suggestion and we have now edited the protocol accordingly to refer to both physical activity and sedentary behaviours under the broader category of movement behaviours.

Reviewer: 2

Dr. Jaclyn P. Maher, Univ N Carolina

Comments to the Author:

This paper is a review protocol and outlines procedures for conducting a review of ecological momentary assessment studies of five distinct health behaviors. My main concern with this protocol is how broad the review is. There are 5 distinct health behaviors which are to be included and the overall objective aim to look at three distinct facets: 1. details of EMA protocol and compliance, 2. associations between psychological factors and behavior (in both directions), and 3. moderators of those associations. I worry about the potential to provide a coherent, digestible review given the broad nature of what is proposed. I have additional, more specific comments outlined below.

Response: We thank the reviewer for their helpful comments on our review protocol. We agree that our review is broad in scope; however, this is intentional. We aim to extract and synthesise key information on EMA protocols (e.g., adherence, incentive schedules) and characteristics of populations and settings across five health-related behaviours, with a view to providing an overview of the field for researchers interested in the application of EMAs to the study of health-related behaviours. We believe such an overarching review would be important for identifying patterns and key knowledge gaps. We have added a clarification of our aims on p.7:

"This review is intentionally broad in scope to provide an overview of the field for researchers interested in the application of EMAs to the study of health-related behaviours. We expect this overarching review to help identify patterns and key knowledge gaps."

We also added to Ethics and Dissemination section (p.15): "We plan to publish overarching review and subsequently five behaviour-specific reviews that will provide a more in-depth synthesis of predictor-behaviour associations"

1. How are the authors defining psychological constructs? At various points through the protocol, they provide examples such as affect, intentions, self-efficacy, but they do not ever seem to actually define the term. Without this it would seem difficult to be able to replicate or determine whether all applicable articles had been included.

Response: We have now added a definition of psychological and contextual variables on p.8: "In this review, we defined psychological variables as emergent properties of a distributed network of neurons, including cognition (e.g., beliefs, attitudes, goals), emotion (e.g., negative affect, cravings) and processes operating on these (e.g., self-regulation, learning), which are linked to behaviour. We further define contextual variables as any potential environmental (e.g., social or physical) influences on behaviour, including the presence of other people, weather, or the availability of unhealthy

foods/cigarettes/alcohol. The psychological and contextual variables will be closely assessed by the reviewers as to their suitability for inclusion/exclusion in the review.”

2. The authors acknowledge that the topic of physical activity will include sedentary behavior, but they have no search terms dedicated to that behavior. Given that physical activity and sedentary behavior are distinct behaviors, it seems as though it would be important to integrate this (and other relevant) search terms.

Response: We agree that this was an important omission and have now added several terms related specifically to sedentary behaviour to our search strategy (e.g. 'sedentary', 'sitting', 'leisure'), described on p.9. The updated search strategy is available in the Supplementary material. See also our responses to Reviewer #1.

3. What is the rationale for including weekly assessments as within the umbrella of EMA? Certainly for physical activity and sedentary behavior, health behaviors which occur multiple times per day, it would seem that weekly assessments would be extrapolating too far for the occasions when these behaviors actually occur (which is the purpose of EMA). Are there certain health behaviors where you would consider weekly assessments to be real time, but others not?

Response: Acknowledging that there is no consensus definition of EMAs, we opted for an inclusive approach. We will include weekly assessments as some of the health-related behaviours of interest in our review might only be executed once per week (e.g. a weekly yoga class, weekly binge drinking, weekly protected/unprotected sexual health behaviours). We therefore considered it an important prerequisite for inclusion in our review that the frequency of the EMAs are appropriate for how the target behaviour (and psychological and contextual predictors) theoretically or empirically unfolds over time. As part of the assessment for inclusion, we will ensure studies that use longer timeframes for data collection i.e. weekly EMAs, have a plausible rationale for this and that the chosen frequency does not reflect the unsuitable aggregation of data. We have clarified this on p.10:

“The frequency of the EMAs should plausibly match how the target behaviour (and psychological and contextual predictors) theoretically or empirically unfolds over time, e.g., daily assessments of steps, weekly assessments of gym class attendance if the class is undertaken only once a week.”

4. The data extraction section doesn't seem to list any information related to objectives 2 or 3. What is listed seems to exclusively focus on describing study characteristics and methods as opposed to the associations examined within these studies.

Response: Thank you - we have now listed additional information for extraction on p.11-12, including statistical model(s) used, level of aggregation in statistical models, coefficients and SEs from statistical models, and control variables in multivariate models. We have also added on p.14 that moderator analyses may be conducted: “Where appropriate, moderator analyses will be conducted to examine whether predictor-behaviour associations vary depending on study setting, study characteristics, participant characteristics, or type of incentive schedule used.”

Reviewer: 3

Dr. Chih-Hsiang Yang, University of South Carolina

Comments to the Author:

This study provides a protocol for reviewing characteristics and features in EMA studies targeting five different health behaviour outcomes. As the EMA literature is growing rapidly in the public health field, synthesizes work is critical and time-sensitive to guide and advance future studies. I consider this proposed work will provide a significant contribution to the field. The study protocol overall is rigorous,

and it complies with current recommendations for conducting systematic reviews.

Response: We would like to thank the reviewer for this positive feedback.

There are some questions and clarifications needed in the current version of the manuscript. I provided my comments and questions for each section for the authors' reference.

Abstract

- Lines 26-28. "Studies need to have assessed at least one psychological or contextual predictor of these behaviours." The inclusion of at least one "contextual predictor" may be too broad for the scope of this review. Contextual factor can also be defined as physical and social contexts, aside from psychological context.

Response: We have now added a definition of psychological and contextual variables considered in this review, please see our response to Reviewer 2, comment 1. We agree that the inclusion of studies including such EMA-assessed contextual variables makes the scope of our review broad. Our intention with this review is to provide a broad overview for researchers interested in health behaviour-related EMA research. We have elaborated further on our aims on p.7:

"The review is intentionally broad in scope to provide an overview of the field for researchers interested in the application of EMAs to the study of health-related behaviours. This overarching review would be important for identifying patterns and key knowledge gaps."

- Lines 31 -33. The authors mentioned that "reporting exclusively on physiological outcomes (e.g., cortisol) or not conducted under free-living conditions will be excluded." Cortisol could also be a psychological outcome to measure stress level. Would it be excluded if a study measures health behaviors and cortisol based on this criterion?

Response: We are not including studies with only physiological outcomes of behaviours. We do, however, include studies that examine physiological measures of psychological variables (e.g., cortisol or heart rate variability to measure stress) together with a measure of at least one of the five health behaviours (please see p.8).

- p.4. Line 16. "length of follow-up", and "assessment type" is unclear. Do the authors mean the time intervals between EMA assessment? And what are some examples for assessment type?

Response: We have now clarified on p.11 that we will extract data on total study duration in days (instead of 'length of follow-up') and EMA method (e.g. signal contingent, event contingent) instead of 'assessment type'.

Introduction

- p.6. A brief literature review/summary of EMA adherence should also be included in the intro.

Response: To our knowledge, our review will be the first to summarise EMA adherence across different health behaviours in adult, non-clinical populations. We have found previous reviews summarising EMA adherence in children/adolescents and specific clinical conditions. We have already cited these reviews in our introduction; however, we would prefer not to elaborate further on the adherence in studies on different populations as the overview is not relevant to the current study.

- p.6. Line 45. "healthy eating" was only mentioned once throughout the MS. Dietary behavior may be a better and more comprehensive term.

Response: Thank you, we have replaced 'healthy eating' with 'dietary behaviour' throughout.

- p.7. Lines 3-5. “predictor-behaviour” and “predictor-outcome” is not clear. Could the authors provide examples to distinguish the two?

Response: We have added an example of a predictor-behaviour association on p.7: “predictor-behaviour” (e.g., stress predicting unhealthy snack consumption).” On reflection, we have decided not to include predictor-outcome associations as our search strategy is not set up to detect studies focused on behavioural outcomes. We have removed any mention of predictor-outcome associations throughout the protocol.

- p.7. Lines 36. Is e-cigarette use included in this review?

Response: We do not focus on e-cigarette use. We have added this clarification in the protocol, page 9. Exclusion criteria: “Studies focusing exclusively on e-cigarettes will be also excluded.”

- p.7. Lines 36. Again, including at least one contextual factor may not help narrow the scope of this review.

Response: We agree - this review is broad in scope. In response to this and previous comments, we have added a definition of contextual variables (see also response 1 to Reviewer 2), and we have also added an explanation that this review is intended to give a broad overview and that it is therefore intentionally broad in scope (see also response to Reviewer 3). Page 7 and page 8.

- p.9. Some critical terms are missing in the search terms. For example, “intraindividual”, “drinking”, “cigarette”, “movement behaviours”, “leisure activity”, “sedentary behaviours”, “addictive behaviour.”

Response: We agree that these terms were missing from our search strategy. We have now added the suggested terms (see Appendix 2),and re-run our searches. The additional terms and suggestions to improve the search strategy have helped us improve the overall quality of the review. Thank you.

Data extraction

- p.10. Line 54. Examples of psychological predictors (e.g., intentions, self-efficacy) should be brought up and define well earlier in the intro as they are key variables in this review.

Response: We have added a definition of psychological variables (see also response 1 to Reviewer 2) and examples of these - page 5 and page 8.

- p.10. Line 26 “A data extraction form will be developed in Microsoft Excel by the two lead authors” and p.11. Line 18 “For each study, one reviewer will extract the data. Twenty percent of studies will be double checked for accuracy and completeness by a second reviewer.” Could you clarify if the two reviewers are the same as the two lead authors?

Response: We have a team of ten authors who are all undertaking data extraction and double checking. We have now clarified this on p.12: “At least 20% of studies stratified by behaviour (e.g., 20% of all alcohol consumption studies) will be double checked for accuracy and completeness by a second reviewer. In case there are any uncertainties related to data extraction (e.g., the primary data extractor is uncertain about a particular parameter or a large number of discrepancies are observed across the primary and secondary data extractor), we will double check additional studies until agreement is achieved. All review authors will be involved in data extraction and double checking.”

- I highly suggest that how the associations between psychological predictors and the behavioural

outcome should also be synthesized. For example, does the study only calculated correlation or use other modeling approaches to test the associations. It will make a valuable contribution to the field.

Response: We agree that this is valuable and have now clarified in the data extraction section that we will also extract data on the type of statistical model implemented (p 14).

- Is there a rationale for only selecting 20% of the studies for double-checking? Some health behavior categories also have more EMA studies than other fields, so the probability for studies being selected for a double check may not be consistent across health behaviours.

Response: We agree with the reviewer's suggestion. We have changed our method and wording to reflect that: "At least 20% of studies stratified by behaviour (e.g., 20% of all alcohol consumption studies) will be double checked for accuracy and completeness by a second reviewer. In case there are any uncertainties related to data extraction (e.g. the primary data extractor is uncertain about a particular parameter or a large number of discrepancies are observed across the primary and secondary data extractor), we will double check additional studies until agreement is achieved. All review authors will be involved in data extraction and double checking."

- p. 13. Lines 22. Standard regression analyses may not be feasible for categorical variables such as setting type, characteristics. Please clarify what analysis will be used for these.

Response: For the analyses pertaining to whether EMA adherence varies depending on study characteristics, participant characteristics, or type of incentive schedule used, the outcome variable will be continuous (i.e. percentage adherence in each study). Therefore, a standard regression-based approach should be sufficient.

Summary

- p. 56. "participant burden" was mentioned the first time in the last sentence of the MS. Study burden is another topic that should not be mixed with study compliance in EMA studies. If reducing burden is another focus of this review to inform future studies, then more discussions are needed in the intro.

Response: We have omitted the words "study burden" from the last sentence of the manuscript - we are not extracting information on the reported burden experienced by participants beyond what can potentially be inferred by EMA adherence rates.

Reviewer: 4

Dr. Saul Shiffman, University of Pittsburgh

Comments to the Author:

This is a reposed protocol for conducting a systematic review of studies of health-relevant behaviors using EMA methods. EMA methods have come into very widespread use in the research areas targeted, and the field could benefit from systematic reviews, so the proposed paper would address a need.

Response: We would like to thank the reviewer for this positive feedback.

The proposal is generally solid and well-thought-out, but there are a few issues that could be better addressed. The search terms seem appropriate, and reflect a knowledge of the variety of traditions in which such research is conducted.

Response: Thank you.

The scope is rather too wide, particularly in trying to encompass studies with multiple within-day measures – the area where innovation has been occurring – along with measures taken only daily or weekly. Especially including weekly measures seems very far outside the scope, and would seem to include studies dating back to the 1800s. At best, this would dilute the review; at worst, the results could be quite muddled. It is recommended the review focus on the more frequent measures that are relatively new.

Response: Thank you - please see our response to reviewer 2, comment 3:

Acknowledging that there is no consensus definition of EMAs, we opted for an inclusive approach. We will include weekly assessments as some of the health-related behaviours of interest in our review might only be executed once per week (e.g. a weekly yoga class, weekly binge drinking, weekly protected/unprotected sexual health behaviours). We therefore considered it an important criterion for our review that the frequency of the EMAs needs to match how the target behaviour (and psychological and contextual predictors) theoretically or empirically unfolds over time. We will assess each behaviour one-by-one and investigate if weekly EMAs match the expected frequency of the target behaviour. We have clarified this on p.7-8:

“The frequency of the EMAs needs to match how the target behaviour (and psychological and contextual predictors) theoretically or empirically unfolds over time, e.g., daily assessments of steps, weekly assessments of gym class attendance if the class is undertaken only once a week.”

We have also added a clarification of our aims on p.7: “This review is intentionally broad in scope to provide an overview of the field for researchers interested in the application of EMAs to the study of health-related behaviours. We expect this overarching review to help identify patterns and key knowledge gaps.”

In another area, the review is too narrow. It proposes to focus only on healthy adults. But a lot of this work is occurring in clinical populations, and it would limit the review to exclude those. Moreover, defining the boundaries is bound to be problematic: are 'problem drinkers' a clinical population or healthy adults with drinking problems? They propose to include studies of obese individuals: Are such individuals 'healthy' or 'clinical'? It may be wise to exclude small highly specialized studies, but not wise to exclude all 'clinical' populations.

Response: As several reviews have already zoomed in on EMA studies conducted in clinical populations (cited in the introduction), we deemed it important and justifiable to focus on 'healthy' (i.e. non-clinical) populations, with a key aim of the review being the comparison across key health-related behaviours. This focus is, to our knowledge, novel and would provide a useful resource with a database of extracted studies published Open Access. As stated on p.9, studies solely recruiting individuals with, e.g., alcohol dependence will be excluded. Studies recruiting 'healthy' individuals with drinking problems would be included as long as recruitment and enrollment procedures are not dependent on a clinical condition or diagnosis. Thus, our procedures allow us to include studies at which individuals with a clinical symptom or condition were able to participate, however, participants were not required to show a clinical symptom or condition.

The 'quality' appraisal needs some revision. It mixes (a) factors that reflect the quality of the DATA (e.g., degree of adherence, which may reflect bias or lack thereof); (b) factors that reflect the quality of REPORTING, such as whether a rationale is stated for the EMA design; and (c) factors that are not 'quality' at all, such as the degree of aggregation, where almost any degree can be appropriate if it fits the data and hypothesis. It seems especially unfortunate to aggregate these diverse considerations into a single index... it could be very misleading.

Response: We fully agree with the reviewer and have decided to move away from a single summary quality indicator. We also agree that the level of data aggregation is not a useful quality indicator and have removed this criterion. These changes are reflected on p.12-13:

“...includes the following four criteria: 1) rationale for EMA design, 2) a-priori power analysis to determine sample size, 3) percentage adherence to the EMA protocol, and 4) treatment of missingness (Table 1).”

“As each criterion refers to a different aspect of study quality, we will not summarise them, but will present how studies score on each selected dimension.”

In sum, the focus and methods seem very sound, but revisions in scope and scoring have the potential to materially improve the resulting review.

Response: Thank you for the positive feedback. We did not deem it necessary to broaden the scope to include clinical populations; however, we have revised our quality appraisal to improve the results of the review.

VERSION 2 – REVIEW

REVIEWER	Maher, Jaclyn P. Univ N Carolina
REVIEW RETURNED	08-Apr-2021

GENERAL COMMENTS	I appreciate the authors efforts in addressing my previous comments. Overall, I am satisfied with their responses. However, I do believe that the quality assessment is missing key elements. For instance, there is no mention of the measures of behavior or psychological constructs in the quality assessment. Using validated measures (or not) would seem to be an important aspect in gauging the quality of the evidence. It seems the authors used the CREMAS to formulate the quality assessment but it is not clear how they settled on the four content areas that are presented in the quality assessment. More details are needed.
--

REVIEWER	Yang, Chih-Hsiang University of South Carolina
REVIEW RETURNED	25-Apr-2021

GENERAL COMMENTS	The authors have properly addressed my previous comments and include suggestions from the reviewers in the revised manuscript. The remaining comment I have is the regression analysis for predicting adherence percentage in each study (page 14). My question was not pertaining to the continuous outcome but the categorical predictors such as study settings, study type, and characteristics. Please provide information for coding these non-continuous variables/predictors, or provide any alternative analytic approaches for testing EMA adherence.
---

VERSION 2 – AUTHOR RESPONSE

Reviewer: 2

Dr. Jaclyn P. Maher, Univ N Carolina

Comments to the Author:

I appreciate the authors efforts in addressing my previous comments. Overall, I am satisfied with their responses. However, I do believe that the quality assessment is missing key elements. For instance, there is no mention of the measures of behavior or psychological constructs in the quality assessment. Using validated measures (or not) would seem to be an important aspect in gauging the quality of the evidence. It seems the authors used the CREMAS to formulate the quality assessment but it is not clear how they settled on the four content areas that are presented in the quality assessment. More details are needed.

Response: Thank you for this suggestion. As the reviewer has highlighted, the quality appraisal tool was developed based on previous literature (i.e. the CREMAS checklist; Stone & Shiffman, 2002) and was also piloted by the systematic review team prior to use. Initially the tool included five criteria; however, after pilot testing and following the protocol review process, one criterion was omitted - i.e., data aggregation as one of the reviewers pointed out that depending on the study design data aggregation may or may not be appropriate. We formed quality criteria using the following CREMAS Items: Rationale, Compliance rate (>80%, 60-79.99%, and <60% - based on Stone & Shiffman, 2002), treatment of missingness (based on Graham, 2009), whether a sample size calculation was included (Stone & Shiffman, 2002) and whether the enrolled sample size met the power analysis indication.

After careful consideration, we decided not to include as part of the final checklist whether the behaviours or psychological predictors were assessed with validated measures. Although we have extracted information regarding the measurement tools used and plan to present these data in the write-up, we do not include them in the quality appraisal. The reason for this decision are two-fold: (1) not all psychological constructs or behaviours can be measured with validated tools and scales; and (2) EMAs often require brief questionnaires and their validity is different to that of full (usually much longer) questionnaires used for one-off comprehensive assessments. In addition, we extracted data on whether there's a 'precedent' for the measure used (which could mean it's been used in a previous study, but no psychometric validation has been conducted), although this information doesn't fit neatly into validated vs. not validated criterion, it will be summarised and elaborated on in the final review.

Reviewer: 3

Dr. Chih-Hsiang Yang, University of South Carolina

Comments to the Author:

The authors have properly addressed my previous comments and include suggestions from the reviewers in the revised manuscript. The remaining comment I have is the regression analysis for predicting adherence percentage in each study (page 14). My question was not pertaining to the continuous outcome but the categorical predictors such as study settings, study type, and characteristics. Please provide information for coding these non-continuous variables/predictors, or provide any alternative analytic approaches for testing EMA adherence.

Response: Thank you for this suggestion. In the article, we state: "To address the third aim, we will assess, with regression analyses, whether EMA adherence varies depending on study setting, study characteristics, participant characteristics, or type of incentive schedule used. We do not have any pre-specified hypotheses. Where appropriate, moderator analyses will be conducted to examine

whether predictor-behaviour associations vary depending on study setting, study characteristics, participant characteristics, or type of incentive schedule used.”

The study predictors were coded (e.g., for study type: observational vs. interventional); for study settings: studies conducted in the US vs. elsewhere (given the majority of studies were conducted in the US); for population: students vs. other (as majority of included studies was conducted with student participants);for EMA characteristics: multiple times per day vs. daily vs. weekly; for incentives: flat payment vs. payment per EMA vs. multiple vs. not reported/no incentive). Currently the above information is described in the data extraction and management section. In the outcome article, these coding and data analysis will be described in further detail.